# Epigenetic regulation of lateralized fetal spinal gene expression underlies hemispheric asymmetries

**Sebastian Ocklenburg[1]\*[†], Judith Schmitz[1][†], Zahra Moinfar[2], Dirk Moser[3], Rena Klose[1], Stephanie Lor[1], Georg Kunz[4], Martin Tegenthoff[5], Pedro Faustmann[2], Clyde Francks[6,7], Jörg T Epplen[8], Robert Kumsta[3], Onur Güntürkün[1,9]**

[1]Institute of Cognitive Neuroscience, Department Biopsychology, Ruhr University Bochum, Bochum, Germany; [2]Department of Neuroanatomy and Molecular Brain Research, Ruhr University Bochum, Bochum, Germany; [3]Department of Genetic Psychology, Ruhr University Bochum, Bochum, Germany; [4]Department of Obstetrics and Gynecology, St. Johannes Hospital, Dortmund, Germany; [5]Department of Neurology, University Hospital Bergmannsheil, Bochum, Germany; [6]Language and Genetics Department, Max Planck Institute for Psycholinguistics, Nijmegen, Netherlands; [7]Donders Institute for Brain, Cognition and Behaviour, Radboud University, Nijmegen, Netherlands; [8]Department of Human Genetics, Ruhr University Bochum, Bochum, Germany; [9]Stellenbosch Institute for Advanced Study (STIAS), Wallenberg Research Centre at Stellenbosch University, Stellenbosch, South Africa

**\*For correspondence:** sebastian.ocklenburg@rub.de

[†]These authors contributed equally to this work

**Competing interests:** The authors declare that no competing interests exist.

**Abstract** Lateralization is a fundamental principle of nervous system organization but its molecular determinants are mostly unknown. In humans, asymmetric gene expression in the fetal cortex has been suggested as the molecular basis of handedness. However, human fetuses already show considerable asymmetries in arm movements before the motor cortex is functionally linked to the spinal cord, making it more likely that spinal gene expression asymmetries form the molecular basis of handedness. We analyzed genome-wide mRNA expression and DNA methylation in cervical and anterior thoracal spinal cord segments of five human fetuses and show development-dependent gene expression asymmetries. These gene expression asymmetries were epigenetically regulated by miRNA expression asymmetries in the TGF-$\beta$ signaling pathway and lateralized methylation of CpG islands. Our findings suggest that molecular mechanisms for epigenetic regulation within the spinal cord constitute the starting point for handedness, implying a fundamental shift in our understanding of the ontogenesis of hemispheric asymmetries in humans.

## Introduction

Compared to the almost infinite complexity of vertebrate cognition and behavior, the number of genes influencing central nervous system development is staggeringly small (*Kadakkuzha and Puthanveettil, 2013*). Thus, understanding the molecular mechanism underlying the epigenetics of vertebrate central nervous system architecture has become an issue central to neuroscience (*Kundakovic and Champagne, 2015*).

One fundamental principle of brain organization is lateralization, i.e. structural or functional difference between the left and the right hemisphere of the brain (*Corballis, 2014*). Lateralization is a

conserved feature across the vertebrate lineage (*Ströckens et al., 2013*; *Ocklenburg et al., 2013a*; *Bisazza et al., 1998*; *Rogers et al., 2012*; *Versace and Vallortigara, 2015*) and recent studies strongly suggest it is also present in invertebrates (*Frasnelli et al., 2012*). This ubiquity of behavioral and brain lateralization strongly supports the idea that lateralized central nervous system organization provides an evolutionary advantage (*Vallortigara and Rogers, 2005*). Suggestions about why a lateralized brain would increase an organism's fitness include avoidance of unnecessary duplication of neuronal activity in both hemispheres, faster neuronal processing due to not being constrained by slow callosal transfer of information between the hemispheres and better coordination of unilateral behaviors in swarms or other social groups of animals (*Vallortigara and Rogers, 2005*; *Corballis, 2009*). In humans, hemispheric asymmetries have been shown in almost all major cognitive systems (*Ocklenburg et al., 2014a*) including language (*Friederici, 2011*; *Sepeta et al., 2016*), memory (*Giammattei and Arndt, 2012*; *Tat and Azuma, 2016*; *Habib et al., 2003*), attention (*Falasca et al., 2015*; *Duecker et al., 2013*), emotional processing (*Demaree et al., 2005*), face perception (*De Winter et al., 2015*), working memory (*Langel et al., 2014*; *Nagel et al., 2013*) and executive functions (*Ocklenburg et al., 2011a*, *2012*; *Stock and Beste, 2014*).

By far the most widely investigated manifestation of lateralization in humans is handedness (*Corballis, 2014*). Importantly, handedness is related to the lateralized organization of cognitive systems in the human brain (*Ocklenburg et al., 2014b*; *Frässle et al., 2016*). For example, left-hemispheric language dominance is found in 96% of right-handed subjects, but only in 73% of left-handed subjects (*Knecht et al., 2000*). The relevance of handedness has recently been highlighted by Willems et al. (*Willems et al., 2014*), who state that it is one of the most important factors influencing the individual brain organization and that explicit inclusion of left-handers in experimental studies has strongly improved our understanding of language, motor behavior and visual processing. Handedness might not only be a behavioral proxy for individual brain organization, but is also interesting from a clinical perspective: A variety of neuropsychiatric and developmental disorders like autism spectrum disorders (*Colby and Parkison, 1977*; *Forrester et al., 2014*; *Preslar et al., 2014*; *Rysstad and Pedersen, 2016*), depression (*Denny, 2009*; *Elias et al., 2001*; *Logue et al., 2015*), bipolar disorder (*van Dyck et al., 2012*; *Nowakowska et al., 2008*), anxiety disorders (*Logue et al., 2015*; *Hicks and Pellegrini, 1978*; *Orme, 1970*; *Wright and Hardie, 2012*; *Hardie et al., 2016*; *Lyle et al., 2013*), schizophrenia (*Hirnstein and Hugdahl, 2014*; *Dragovic and Hammond, 2005*; *Sommer et al., 2001*) or alcoholism (*Denny, 2011*; *Mandal et al., 2000*; *Sperling et al., 2000*) has been associated with left- and mixed-handedness. Thus, understanding the ontogenesis of handedness and hemispheric asymmetries in general could potentially yield important insights into pathogenesis of these disorders.

However, despite their importance for many aspects of brain organization, the ontogenetic background of brain asymmetries is still far from being understood. Initially, single gene theories have been suggested to explain the emergence of handedness as a function of one gene with two alleles (*Annett, 1998*; *McManus, 1985*). However, recent genome wide association studies failed to detect any genome-wide significant single nucleotide polymorphisms, refuting single gene theories (*Armour et al., 2014*; *Eriksson et al., 2010*). Candidate gene studies revealed a number of genes that display an association with handedness, among them *leucine rich repeat transmembrane neuronal 1 (LRRTM1)* (*Francks et al., 2007*), *proprotein convertase subtilisin/kexin type 6 (PCSK6)* (*Scerri et al., 2011*; *Arning et al., 2013*; *Brandler et al., 2013*) and the *androgen receptor gene (AR)* (*Arning et al., 2015*; *Hampson and Sankar, 2012*; *Medland et al., 2005*). However, these genes explain only a fraction of the variance in handedness data. Moreover, a number of studies has suggested that only about one quarter of the variance in handedness is attributed to genetic variation, whereas the remaining 75% of variance are explained by non-shared environmental factors (*Medland et al., 2006*, *2009*; *Vuoksimaa et al., 2009*). These findings highlight the importance of integrating both genetic variation and epigenetic processes modulating gene expression when investigating the ontogenesis of hemispheric asymmetries (*Geschwind and Miller, 2001*).

Asymmetric gene expression in the fetal cortex has been suggested as the molecular basis of left-right differences in hand-use: Sun et al. (*Sun et al., 2005*) compared gene expression levels in the right and left perisylvian cortex of the human fetus. At 12 gestational weeks, the authors identified 27 consistently asymmetrically expressed genes, which are mostly responsible for gene expression regulation, signal transduction, and cortical development. One of the consistently asymmetrically expressed genes was *LIM Domain Only 4 (LMO4)*. Further investigation revealed that unilateral

variation of *Lmo4* expression in embryonic mice suppresses neurogenesis in one hemisphere, leading to the asymmetric functional area formation, neuronal production and axonal projection as well as altered paw preference (*Li et al., 2013*). Analysis of gene expression in the adult human brain yielded less clear results, since two independent studies found no differences in gene expression between analogous regions across the cerebral hemispheres (*Hawrylycz et al., 2012*; *Pletikos et al., 2014*). In a recent study, Karlebach and Francks reanalyzed both datasets and showed that subtle lateralization at single gene level translates to stronger asymmetries at the level of functional gene ontology (GO) groups. The authors found lateralized gene sets to be associated with neuronal electrophysiology, synaptic transmission, nervous system development, and glutamate receptor activity (*Karlebach and Francks, 2015*).

However, recent research indicates that cortical tissue might not be the optimal choice to investigate the relation of gene expression asymmetries and behavioral asymmetries. Ontogenetically, handedness starts early in development since coordinated hand movements begin 8 weeks post conception (PC), i.e. 10 weeks gestational age, when 85% of fetuses exhibit more right arm than left arm movements (*Hepper et al., 1998*; *de Vries et al., 1985*). Investigation of thumb sucking in 274 fetuses revealed that at 13 weeks PC 90% prefer to suck their right thumb whereas only 10% suck their left thumb more often (*Hepper et al., 1990*, *1991*). Interestingly, a follow up study of 75 infants revealed that thumb sucking preference is significantly positively correlated with subsequent handedness: The 60 children showing a right thumb preference were right-handed whereas out of the 15 children displaying a left thumb preference, five were right-handed and 10 were left-handed (*Hepper et al., 2005*). Importantly, the motor cortex is not yet functionally linked to the spinal cord at that stage of development as the outgrowth of corticospinal projections does not enter the anterior spinal cord before 15 weeks PC (*ten Donkelaar et al., 2004*). This implies that handedness is unlikely to be under brain control (*Hepper et al., 1991*) and asymmetrical hand movements have to arise from spinal activity patterns. Thus, it is likely that spinal rather than cortical gene expression asymmetries represent the molecular basis of handedness.

Asymmetrical gene expression patterns are likely to be influenced by epigenetic variation. The most important epigenetic mechanism is DNA methylation. Binding of methyl (-CH$_3$) groups to CpG sites or islands causes a reduction or prevention of transcription and thus gene expression. DNA methylation is confirmed to be involved in the development of basic central nervous system functions like synaptic function, neuronal plasticity, learning and memory (*Nikolova and Hariri, 2015*; *Day et al., 2015*; *Roth, 2012*). Especially intrauterine stressors have been shown to influence DNA methylation (*Turecki and Meaney, 2016*; *Vaiserman, 2015*), which is particularly interesting in the context of handedness ontogenesis. Moreover, a recent study showed that methylation plays a role in the ontogenesis of handedness: methylation levels in a CpG block in the promoter region of *LRRTM1* were associated with atypical handedness (*Leach et al., 2014*).

Post-transcriptionally, gene expression is further regulated by microRNAs (miRNAs) that are composed of small, 21–25 nucleotide, non-coding RNAs. In humans and other mammals, miRNAs primarily cause destabilization of target mRNAs instead of reduced translation (*Guo et al., 2010*). This has also been shown to be relevant for hemispheric asymmetries, as neuronal asymmetries in the nematode *Caenorhabditis elegans* are controlled for by different miRNAs (*Alqadah et al., 2013*; *Cochella and Hobert, 2012*; *Johnston and Hobert, 2003*; *Hsieh et al., 2012*).

Pronounced changes in spatiotemporal expression profiles are a key feature of human embryogenesis (*Yi et al., 2010*) and formation of functional asymmetries in vertebrates has been shown to strongly depend on critical periods in early development (*Le Grand et al., 2003*; *Zappia and Rogers, 1983*). To investigate the molecular determinants of human behavioral asymmetries we analyzed asymmetries in genome-wide mRNA expression, miRNA expression and DNA methylation patterns in human fetal spinal cord tissue. Importantly, we specifically wanted to investigate the spinal cord segments innervating arms and hands. While rostral cervical segments (C2–C5) innervate the head, neck and shoulder region, the subsequent segments directly innervate arms and hands with C6 innervating the thumb, C7 innervating the middle finger and C8 innervating the little finger. T1 innervates the medial site of the antecubital fossa (*Maynard et al., 1997*). Based on the findings on the start of left-right asymmetries in coordinated hand movements (*Hepper et al., 1998*; *de Vries et al., 1985*), we focused on fetal tissue samples obtained between 8 and 12 weeks PC.

We hypothesized that gene expression asymmetries between the left and right spinal cord start at 8 weeks PC, as this is the starting point of coordinated asymmetrical hand movements. Based on the findings about the role of non-genetic influence factors for handedness development, we also assumed a pronounced modulation of these mRNA expression asymmetries by asymmetric DNA methylation and asymmetric miRNA expression.

## Results

### Gene expression

Asymmetries in mRNA expression in spinal cord segments C2 to T2 were observed at all three developmental stages, with the largest differences evident at 8 weeks PC. At 8 weeks PC, 1690 transcripts (3.29%) showed left-right gene expression differences with $\log_2$(fold change) > 1.5. The fact that 39 transcripts showed stronger left-sided gene expression, while 1651 transcripts showed stronger right-sided gene expression highlights increased right-sided gene expression in the spinal cord at this developmental stage. The number of asymmetrically expressed genes with a $\log_2$(fold change) > 1.5 was reduced to only 24 genes (0.05%) at 10 weeks PC. Among these, 15 displayed leftward asymmetrical gene expression and nine showed rightward asymmetrical gene expression. Four genes (0.01%) showed a $\log_2$(fold change) of 1.5 or higher (see *Figure 1* for top 25 asymmetrically expressed genes per developmental stage and *Supplementary file 1G* for individual samples) at 12 weeks PC, all indicating stronger gene expression in the right spinal cord. Among the candidate genes associated with the development of hemispheric asymmetries (see *Figure 2A*), *forkhead box P2 (FOXP2)* (*Ocklenburg et al., 2013b*) displayed a rightward asymmetry in the spinal cord at 10 weeks PC. *BDNF antisense RNA (BDNF-AS)* (*Manns et al., 2008*) was higher expressed in the left spinal cord at 8 weeks PC.

### GO groups

Recently, it has been suggested that subtle expression asymmetries at the level of individual genes may translate to stronger asymmetries within the gene ontology (GO) groups (*Karlebach and Francks, 2015*). The number of significant GO groups (p<0.05) as displayed by enrichment analysis over asymmetrically expressed genes per hemisphere and developmental stage was the highest at 8 weeks PC (123 overall). For the 69 transcripts showing asymmetrical gene expression towards the left spinal cord, WebGestalt revealed three enriched GO groups: Platelet-derived growth factor binding (p<0.05), collagen (p<0.05), and fibrillary collagen (p<0.05). In contrast, for the 1651 transcripts showing asymmetrical gene expression towards the right spinal cord, GO analysis revealed 120 enriched GO groups displaying particular involvement in biological processes like cell cycle (p<0.001), cellular component organization or biogenesis (p<0.001), and metabolic processes (p<0.001). Enriched molecular functions include protein binding (p<0.001), transferase activity (p<0.001), and protein binding transcription factor activity (p<0.001). The number of significant GO groups was reduced at 10 weeks PC (41 overall): The 15 genes showing leftward asymmetric gene expression cluster in 4 GO categories representing cellular components: DNA-directed RNA polymerase II, holoenzyme (p<0.05), nuclear DNA-directed RNA polymerase complex (p<0.05), RNA polymerase complex (p<0.05), and DNA-directed RNA polymerase complex (p<0.05). GO categories enriched in the nine genes asymmetrically expressed towards the right spinal cord include system development (p<0.05), regulation of reproductive process (p<0.05), cell proliferation (p<0.05), and multicellular organismal process (p<0.05). At 12 weeks PC, no GO group reached statistical significance (see *Figure 2B–C* and related *Figure 2—source data 1*).

### miRNA

At 8 weeks PC, 301 miRNA transcripts were expressed in both the left and the right spinal cord. Out of those, five (1.66%) showed a biologically relevant asymmetry towards the right spinal cord. At 10 weeks PC, six of 382 transcripts (1.57%) displayed a $\log_2$(fold change) > 1.5, thereof three leftwards and three rightwards. At 12 weeks PC, seven of 294 expressed transcripts (2.38%) were differentially expressed with six being more strongly expressed in the left spinal cord and one being more strongly expressed in the right spinal cord (see *Figure 3A* and related *Figure 3—source data 1*). For each developmental stage, target genes of asymmetrically expressed miRNA transcripts were

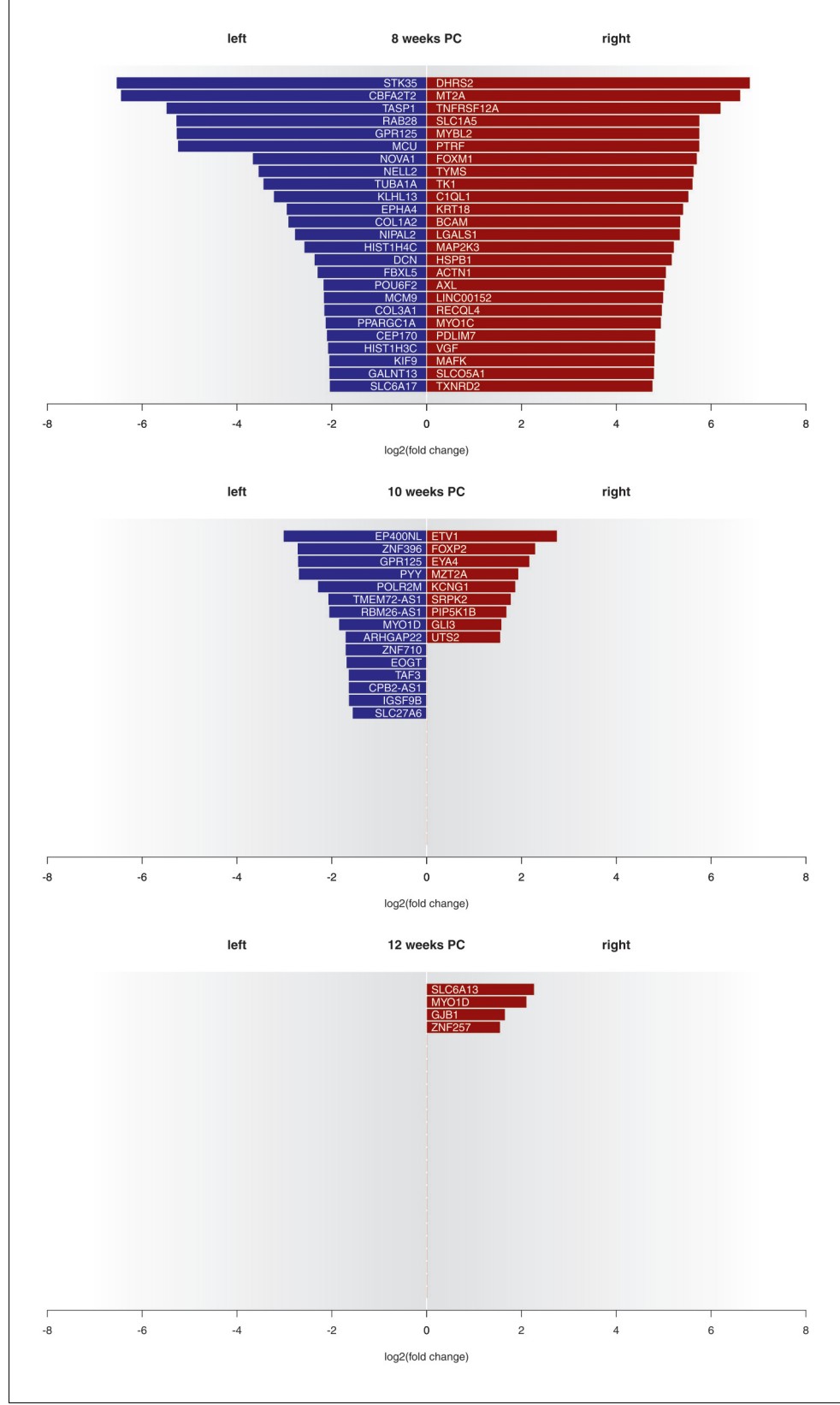

**Figure 1.** Gene expression asymmetries in human fetal spinal cord at 8, 10 and 12 weeks PC. X-axis shows the extent of asymmetry measured in log₂(fold change) between right and left spinal cord samples. Blue bars show leftward asymmetrically expressed genes, red bars show rightward asymmetrically expressed genes. For 8 weeks

*Figure 1 continued*

PC, the top 25 genes with highest rightward/leftward gene expression asymmetries are depicted. For 10 and 12 weeks PC, all genes with a log$_2$(fold change) > 1.5 are shown. The source files of asymmetrically expressed genes per developmental stage with corresponding fold change values are available in *Figure 1—source data 1*.
The following source data is available for figure 1:

**Source data 1.** Asymmetrically expressed genes per week.

compared to asymmetrically expressed genes. At 8 weeks PC, 65 of the 1690 asymmetrically expressed genes (3.85%) were likely to be targets of asymmetrically expressed miRNA transcripts. At 10 weeks PC, six of the 24 asymmetrically expressed genes (25%) were targets of differentially expressed miRNAs. At 12 weeks PC, one of the four asymmetrically expressed genes (25%) was a target of the asymmetrically expressed miRNAs of that developmental stage. For the miRNAs asymmetrically expressed towards the right spinal cord at 8 weeks PC, 12 KEGG pathways reached FDR-corrected significance. By far the largest effect was observed for the Transforming growth factor beta (TGF-$\beta$) signaling pathway (p<0.001), which all five miRNAs asymmetrically expressed towards the right spinal cord were involved in. Additionally, among the 10 genes involved in this pathway, two (*SP1, SMAD3*) were differentially expressed at 8 weeks PC. At 10 weeks PC, four pathways reached FDR-corrected significance (two left, two right). For 12 weeks PC, 16 pathways were detected in KEGG analyses (14 left, two right).

## DNA methylation

At 8 weeks PC, 31,278 CpG sites showed higher DNA methylation (FDR-corrected p-value below 0.01 and the % methylation difference between left and right above 25%) in the left spinal cord over both samples, whereas only 8615 CpG sites were more extensively methylated in the right spinal cord of both samples. At 10 weeks PC, 10,892 CpG sites showed side-specific asymmetrical DNA methylation towards the left side and 11,081 towards the right side of both samples (see *Figure 3B–C* and related *Figure 3—source data 4*). At 12 weeks PC, for which only one sample was available, 281,119 CpG sites showed higher DNA methylation in the left and 352,118 in the right spinal cord. Comparing the methylation data to gene expression data revealed that at 8 weeks PC, 451 of 1690 asymmetrically expressed genes were asymmetrically methylated towards the opposite direction, thus, 27% of the variance in asymmetrical gene expression could be explained by differential methylation alone. Moreover, 1% of variance (18 genes) could be explained by the asymmetrical miRNA expression as well as differential methylation and 3% of variance (47 genes) could be explained by miRNA alone, which leaves 69% of variance unexplained (*Figure 3D*). At 10 weeks PC, 25% of variance in asymmetrical gene expression (six genes) could be explained by miRNA alone, followed by methylation (8%, two genes). 67% of variance remained unexplained. At 12 weeks PC, 25% of variance (one gene) was explained by miRNA and 25% (one gene) by methylation. 50% of variance remained unexplained at 12 weeks PC.

## Discussion

Hemispheric asymmetries in brain and behavior are a major organizational principle in the vertebrate central nervous system, but their ontogenesis is not well understood (*Ocklenburg et al., 2014b*). While it is general consensus that both genetic and epigenetic factors play a role (*Ocklenburg et al., 2013c*), it is unclear, which molecular processes underlie the epigenetic modulation of gene expression asymmetries, a potential origin of behavioral asymmetries (*Sun et al., 2005*; *Karlebach and Francks, 2015*). To elucidate this question we analyzed asymmetries in genome-wide mRNA expression, miRNA expression and DNA methylation patterns in human fetal tissue samples. Importantly, we focused on spinal cord, not brain, tissue. Eight weeks after conception, human fetuses exhibit pronounced lateralized motor behavior of the arms. As cortical control of this behavior is unlikely (*Hepper et al., 1998*; *de Vries et al., 1985*; *Hepper et al., 2005*; *ten Donkelaar et al., 2004*), it has been suggested that it is under spinal control (*Hepper, 2013*).

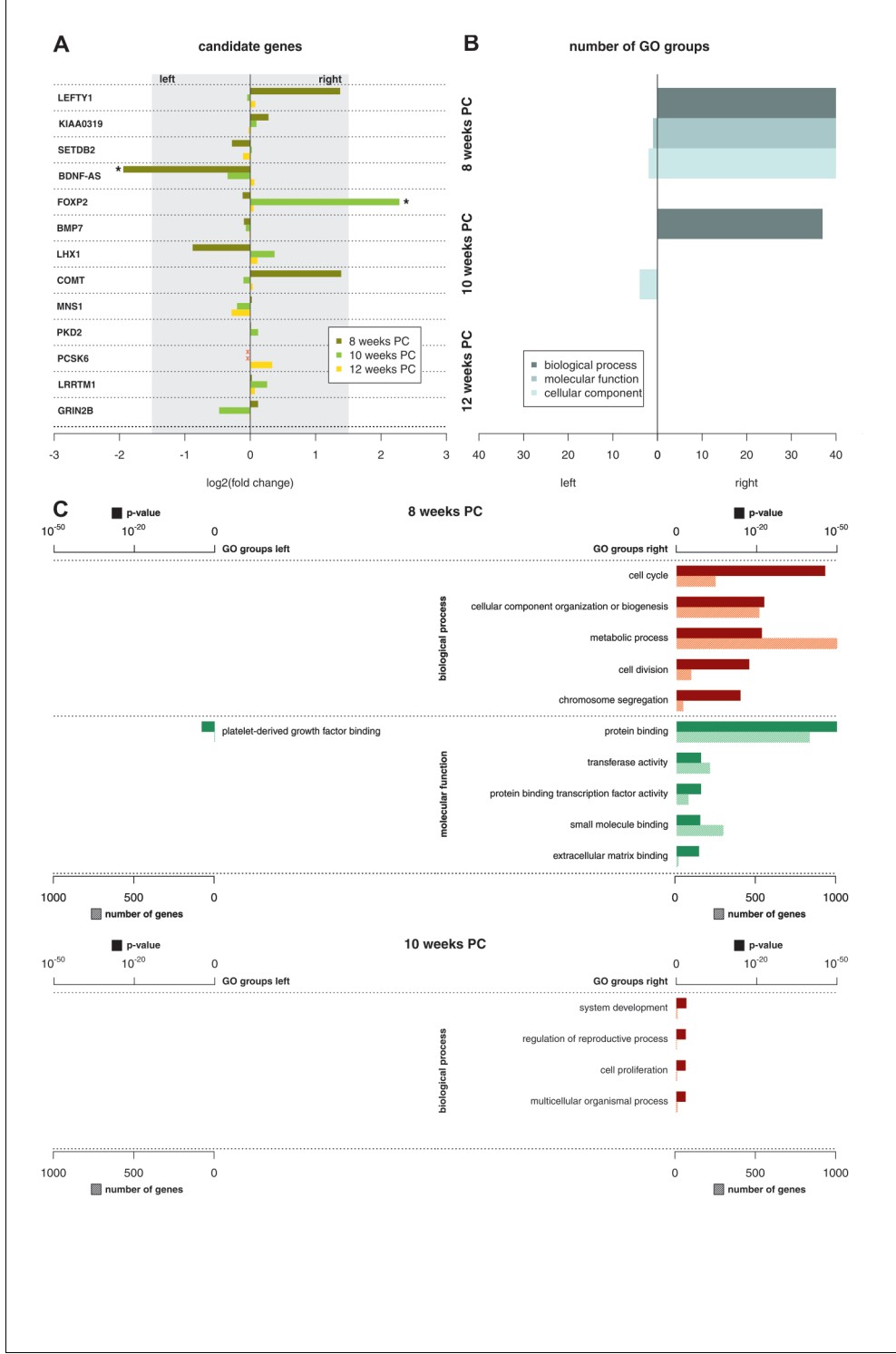

**Figure 2.** Functional genes and gene groups. (**A**) Gene expression asymmetries for previously published candidate genes for handedness and functional lateralization. Asterisks indicate biologically relevant gene expression asymmetry with a log₂(fold change) > 1.5. (**B**) Number of significant Gene Ontology (GO) groups for the three main categories 'biological processes', 'molecular function' and 'cellular component' for weeks 8, 10 and 12 PC. (**C**) Main GO groups for 8 and 10 weeks PC with p-value and number of involved genes for the left and right spinal cord. The source files of all enriched GO groups are available in *Figure 2—source data 1*.

The following source data is available for figure 2:

*Figure 2 continued on next page*

*Figure 2 continued*

**Source data 1.** Enriched GO groups per week and side of the spinal cord.

In line with our hypothesis, our findings suggest that gene expression asymmetries in the spinal cord segments innervating the hands and arms might be critical for the ontogenesis of functional asymmetries. For the first time we show that the left and right cervical and anterior thoracic segments of the fetal spinal cord do show biologically relevant gene expression differences. Importantly, these gene expression asymmetries are highly developmental stage-specific. At eight weeks after conception, gene expression asymmetries between the two halves of the spinal cord were most pronounced, with 3.29% of all transcripts showing biologically relevant left-right gene expression differences, largely higher towards the right side and involved in numerous GO categories contributing to neurodevelopment. At 10 weeks PC, this number decreased substantially to 0.05% and further so at 12 weeks PC (0.01%). While the findings for 10 and 12 weeks PC are largely comparable to what has been reported for gene expression asymmetries in the fetal cerebral cortex (*Sun et al., 2005*; *Karlebach and Francks, 2015*), the data for eight weeks PC indicate a substantial increase over previous reports of gene expression asymmetries in CNS tissue that goes along with the first onset of coordinated hand movements.

In line with the suggestion that non-shared environmental influences account for more than 75% of the variance in functional hemispheric asymmetries in humans (*Medland et al., 2009*), we could show that a large part of these gene expression asymmetries is regulated by epigenetic processes.

On the one hand, we could show that DNA methylation of CpG islands shows substantial asymmetries that are related to RNA expression asymmetries. In week 8 tissue samples, there was a strong left-lateralization of CpG island methylation, indicating a stronger repression of gene transcription in the left spinal cord. This is well in line with our finding of increased overall right-sided gene expression at that time point. Direct comparison of the location of asymmetrically methylated CpG islands and asymmetrically expressed genes indicated that 27% of the variance in asymmetrical gene expression at week 8 could be explained by differential methylation. In week 10 tissue samples, methylation asymmetries are massively decreased as compared to week eight, also in line with the gene expression data. Week 12 is difficult to interpret as here only one sample was analyzed, greatly increasing the number of asymmetrically methylated CpG sites.

On the other hand, we could also show that the asymmetries in gene expression are modulated by miRNA expression asymmetries. Particularly interesting was our finding that for the miRNAs asymmetrically expressed towards the right spinal cord at week eight, KEGG pathway analysis revealed a substantial effect of the TGF-$\beta$ signaling pathway. This is an intriguing finding, as both nodal growth differentiation factor (Nodal) and left-right determination factor (Lefty), two of the key proteins for establishing bodily left–right asymmetry during development are part of the TGF-$\beta$ superfamily (*Mittwoch, 2008*; *Shiratori and Hamada, 2014*). Importantly, TGF-$\beta$ signaling has directly been linked to handedness, as *proprotein convertase subtilisin/kexin type 6 (PCSK6)*, one of the major candidate genes for handedness, encodes for a protease that cleaves NODAL (*Scerri et al., 2011*).

Our data collection was limited to weeks 8 to 12 PC and for future studies, it could potentially be interesting to include tissue samples from even earlier stages to get a more detailed picture of the developmental trajectory. Also, independent replication in larger samples is needed in order to make more in-depth functional conclusions. As limb preferences have been reported in many non-human primates, but the strong 90 to 10 distribution of right- and left-handedness in humans seems to be rather unique, comparative analysis of primate tissue samples might yield unique insights into the evolution of the molecular basis of hemispheric asymmetries.

One potential issue with the interpretation of our data is to what extent the observed gene expression asymmetries are linked to visceral situs and not necessarily nervous system asymmetries. A particular interesting experiment in this regard would be to investigate spinal cord gene expression asymmetries in the inversus viscerum (iv) line of mouse mutants (*Okada et al., 1999*). These mutants show randomized visceral laterality and by comparing spinal cord gene expression asymmetries between animals with normal and atypical visceral asymmetries a potential impact of visceral asymmetries on spinal cord gene expression could easily be identified.

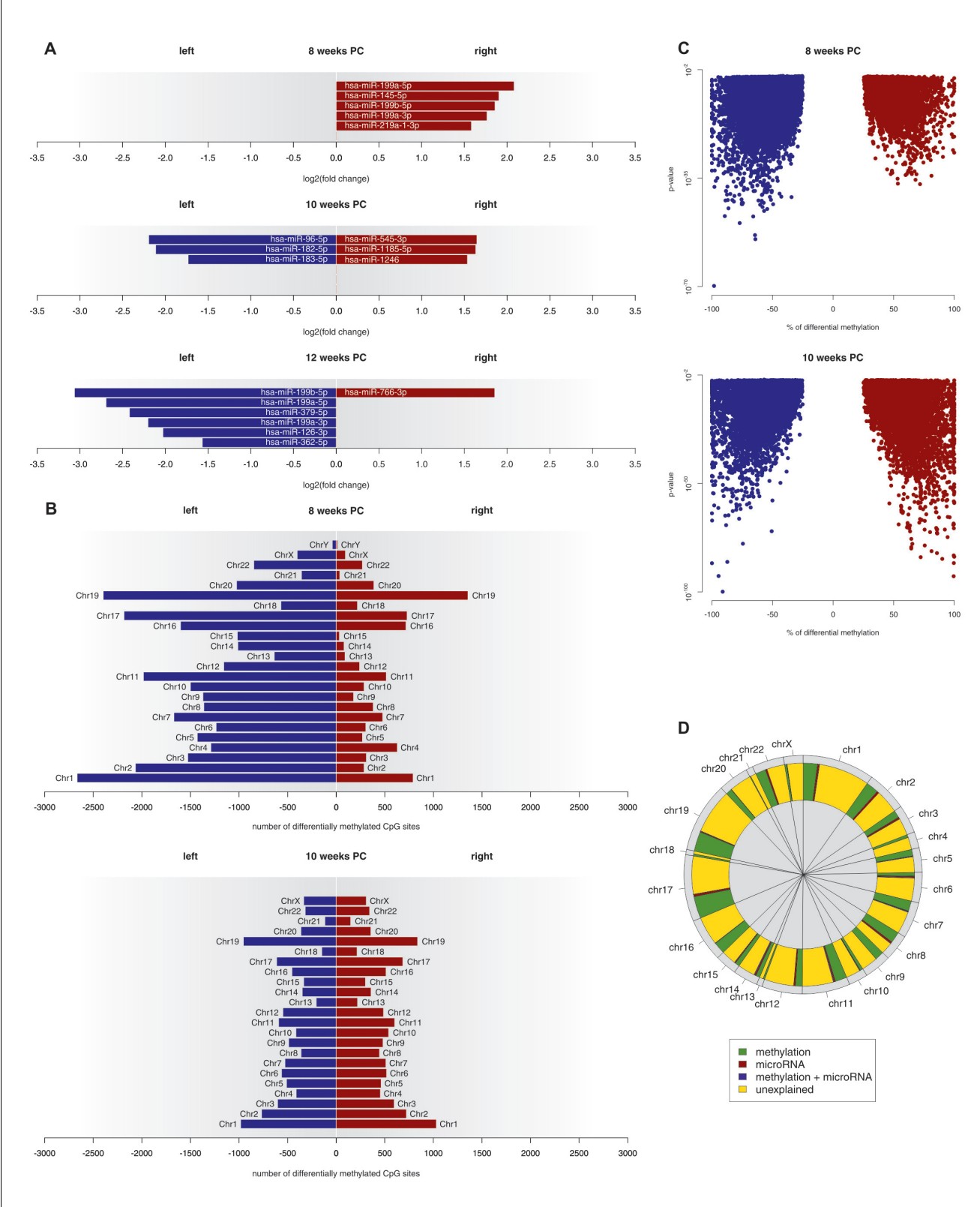

**Figure 3.** Epigenetic regulation of gene expression asymmetries in human fetal spinal cord. (**A**) Asymmetrically expressed miRNA transcripts at 8, 10 and 12 weeks PC. The extent of expression asymmetries is measured in $\log_2$(fold change). Red bars show rightward asymmetrically expressed microRNA transcripts, blue bars show leftward asymmetrically expressed miRNA transcripts. (**B**) Number of CpG sites showing differential DNA methylation per chromosome, compared between the left and right spinal cord for 8 and 10 weeks PC. Depicted are only CpG sites with methylation

*Figure 3 continued on next page*

*Figure 3 continued*

asymmetries in both samples. Red bars represent the number of CpG sites that showed significantly higher DNA methylation on the right side, blue bars show the number of CpG sites that showed significantly more DNA methylation on the left side. (C) Percentage of differential DNA methylation in leftward (blue) and rightward (red) asymmetrically methylated CpG sites as a function of p-value. (D) Percentage of gene expression asymmetries on each chromosome at 8 weeks PC that can be explained by regulation via asymmetrically expressed miRNAs or asymmetric DNA methylation of CpG sites within and 1500 nucleotides upstream of the expressed genes. The source files of asymmetrically expressed miRNAs, asymmetrically expressed targets of miRNAs, enriched KEGG pathways and differentially methylated CpG sites are available in *Figure 3—source data 1*, *Figure 3—source data 2*, *Figure 3—source data 3*, and *Figure 3—source data 4* respectively.

The following source data is available for figure 3:

**Source data 1.** Asymmetrically expressed miRNAs per week.

**Source data 2.** Asymmetrically expressed RNA targets of asymmetrically expressed miRNAs per week.

**Source data 3.** Enriched KEGG pathways per week and side of the spinal cord.

**Source data 4.** Asymmetrically methylated CpG sites per week and side of the spinal cord.

Taken together, these results are a strong indicator of epigenetic influences on human spinal cord gene expression asymmetries, a potential precursor of handedness. In birds, it has been shown that a behavioral preference for turning the head to the right, caused by epigenetic modulation during a critical period just before hatch, induces not only motor, but also visual and cognitive asymmetries (*Casey and Martino, 2000*; *Manns and Güntürkün, 2009*; *Rogers, 1982*; *Skiba et al., 2002*). Based on our data, a similar model is conceivable in humans. As week eight after conception represents the onset of coordinated hand movements and behavioral asymmetries of the hands occur first at this time point (*Hepper et al., 1998*; *de Vries et al., 1985*), we assume that a certain time frame before 10 weeks PC represents the critical period for handedness formation. During this period, asymmetrical DNA methylation and posttranscriptional regulation by asymmetrically expressed miRNAs lead to a spike in RNA expression asymmetries in the spinal segments innervating the arms and hands. These expression asymmetries of genes relevant for CNS development could lead to a differential development of neuronal circuits influencing the right arm and hand, causing the described behavioral asymmetries. For example, it has been shown that spinal cord segments innervating the right arm contain motoneurons with larger somata than left arm segments. In contrast, there are no size differences in left and right segments innervating the upper trunk (*Melsbach et al., 1996*). Furthermore, at eight weeks PC, the spinal cord and motor cortex are functionally not connected (*ten Donkelaar et al., 2004*), also supporting that behavioral asymmetries in arm use at that stage are controlled for by the spinal cord. At a later developmental stage when the spinal cord and motor cortex are functionally connected, the established behavioral asymmetry then could lead to asymmetries in use-dependent neuronal plasticity processes (*Cirillo et al., 2010*) in the motor cortex, ultimately leading to the cortical correlates of handedness (*Ocklenburg et al., 2013c*). This process could start at around 14 weeks PC, as asymmetric fetal hand use at that stage strongly correlates with later handedness at school age (*Hepper et al., 1990*, *2005*). Unlike models assuming that handedness is primarily controlled by allelic variations in one or more candidate genes (*Annett, 1998*), our suggestion is in line with the finding that more than 75% of the variance in handedness data is explained for by non-shared environmental influences (*Medland et al., 2009*). Moreover, our data do not contradict neither linkage studies in extended left-handed pedigrees nor genome-wide association studies which were unable to identify allelic variants that explain more than a fraction of the variance in handedness data (*Armour et al., 2014*; *Eriksson et al., 2010*; *Somers et al., 2015*). Whereas our findings suggest that a large part of these influencing factors act prenatally, there remain several important peri- and postnatal environmental factors like social modulation to shape actual handedness (*Schaafsma et al., 2009*).

In summary, we could show pronounced, time sensitive gene expression asymmetries in human fetal spinal tissue that overlap with the onset of behavioral asymmetries. Thus, our data suggest a spinal, not a cortical, beginning of hemispheric asymmetries. The observed gene expression asymmetries were modulated by asymmetric CpG island methylation and asymmetries in miRNA

expression, suggesting that these processes form the molecular basis of asymmetry epigenetics. In conclusion, our data strongly suggest a multifactorial model for the ontogenesis of hemispheric asymmetries, including both multiple genetic and epigenetic factors.

## Materials and methods

### Sample collection

The human spinal cord was collected from fetal tissue discarded following induced pregnancy termination in a regional gynecology clinic. None of the physicians or other medical personnel involved in conducting the pregnancy terminations was involved in this scientific study. The study was approved by the Ethics Committee of the Medical Faculty of the Ruhr University Bochum (registration number 5056–14). All fetal tissue donors signed written informed consent at least 24 hr before the pregnancy termination was conducted. Following informed consent, the handedness of fetal tissue donors was determined using the Edinburgh Handedness Inventory (EHI) (*Oldfield, 1971*).

### Tissue preparation

Tissue samples were dissected from the spinal cord of six fetuses after pregnancy terminations at 8, 10, and 12 weeks post conception (PC), i.e. 10, 12 and 14 weeks gestational age. Due to ethical considerations when working with aborted human fetal tissue, the sample size was limited to six fetuses for which we got allowance by the Ethics Committee. This number was based on effects in previous studies with fetal cortical tissue (*Sun et al., 2005*). Fetal pathologies were ruled out as far as possible by excluding pregnancy terminations due to medical indications as well as karyotype aberrations. The samples were also excluded in case of heavily destructed tissue. Following pregnancy termination, fetal and surrounding tissue was immediately rinsed with sterile 1x phosphate buffered saline (PBS) in order to hold ion concentrations constant while blood was removed. In case the spine was detectable, it was fixed with sterile cannulas with a diameter of 0.6 mm (B Braun, Melsungen, Germany) and opened longitudinally with ball-ended dissecting scissors. In order to differentiate left and right, the right spinal cord was marked with small injections of 1% cresyl violet. Tissue samples were stored in 1 ml Allprotect Tissue Reagent (Qiagen, Hilden, Germany). Spinal cord tissue preparation was conducted as quickly as possible to ensure that RNA was not degraded (7:50–16:40 min, see *Supplementary file 1A*). Subsequently, 50–100 mg of chorionic villi were removed and stored in 10 ml of RPMI 1640 Medium (Life Technologies GmbH, Carlsbad, California) for subsequent karyogram analysis. All tissue samples were transported to Ruhr University Bochum, Germany. Spinal cord samples were stored at 4°C to preserve the gene expression profile. 24 hr later, the upper third of the spinal cord was separated on a Teflon freezing plate in order to include spinal cord segments C2 to T2. The left and right spinal cord were dissected by separating the tissue longitudinally along the midline and restored in the Allprotect Tissue Reagent at −80°C.

### Karyotyping

Karyograms were assembled at the Department of Human Genetics (Ruhr University Bochum) to ensure that karyotypes were normal without major chromosomal aberrations. Cell cultures were incubated with 1 μg colcemid (Gibco, Karlsruhe, Germany) for 80 min and were then harvested from flask applying trypsin-EDTA (0.05/0.02 w/v) (Biochrome, Berlin, Germany) for 2–4 min. After transferring to tube and centrifugation (170 g for 10 min) cells were incubated in 0.56% KCl hypotonic solution for 20 min and subsequently fixed and washed using a 3:1 methanol–glacial acetic acid fixative. After spreading on slides and air drying samples were stained in 0.025% quinacrine hydrochloride (Sigma, Steinheim, Germany) for 20 min. Q bands were visualized on a Zeiss Axioskop 2 fluorescent microscope and 100 metaphases per sample were analyzed in Ikaros software (Metasystems, Altlussheim, Germany).

### Assessment of RNA and DNA

Total RNA including miRNA and DNA was extracted using the AllPrep DNA/RNA/miRNA Universal Kit (Qiagen, Hilden, Germany) according to the manufacturer's instructions. Concentration and purity of RNA and DNA were determined photometrically (NanoDrop ND-1000 Spectrophotometer, Thermo Scientific, Waltham, Massachusetts). RNA quality was controlled using the Agilent RNA

6000 Pico Kit and Agilent 2100 Bioanalyzer (Agilent Technologies, Santa Clara, California) according to the manufacturer's recommendations. Extracted RNA and DNA were stored at −80˚C until gene expression analysis was performed. For RNA and DNA quality measurements see *Supplementary files 1B–C*.

## Gene expression analysis and bioinformatics

'INVIEW Transcriptome Discover' provided by GATC Biotech AG (Konstanz, Germany) was used to analyze the extracted mRNA. Sample VI did not pass entry quality control and was not further processed. For the remaining samples, rRNA was depleted from total RNA for purification and subsequent fragmentation of mRNA into RNA-Seq reads. A strand-specific cDNA-library was generated for subsequent Illumina paired-end sequencing with 60 million reads. The RNA-Seq reads were aligned to the reference genome (*Homo sapiens*, hg19) using Bowtie (RRID:SCR_005476). Read statistics are reported in *Supplementary files 1D–E*. Potential exon-exon splice junctions were discovered (TopHat, RRID:SCR_013035). The software Cufflinks (RRID:SCR_013307) (*Trapnell et al., 2010*) then recognized and quantified transcripts, which were merged to full length transcripts and annotated. Cuffdiff (RRID:SCR_001647) tracked the mapped reads and determined the relative gene expression value (fragment per kilo base of transcript per million fragments mapped [FPKM]) for each transcript in each sample.

Overall, gene expression was investigated in 51,408 transcripts in left and right spinal cord of five samples. Asymmetric gene expression was determined for annotated genes, which were identified using the RefSeq database (http://www.ncbi.nlm.nih.gov/refseq/; RRID:SCR_003496). Differential gene expression is usually reported as a fold change in FPKM. In case of very small FPKM values the fold change is likely to be high although the difference in gene expression is small (*Warden et al., 2013*), so a gene was considered as abundant if FPKM was at least 1. The number of protein coding genes with an FPKM-threshold of 1 on both sides was slightly lower in sample II (9102, 17.7%) than in the other samples (I: 10,618, 20.7%, III: 10,970, 21.3%, IV: 10,854, 21.1%, V: 10,989, 21.4%). Due to the small sample size, differential gene expression for 8 and 10 weeks PC was determined by fold change of means using a threshold of $\log_2$(fold change) > 1.5. This value is commonly acknowledged to indicate a gene expression difference with possible functional relevance (*Hawrylycz et al., 2012*). At 12 weeks PC, only one sample was included and genes with a $\log_2$(fold change) > 1.5 of FPKM values were considered as asymmetrically expressed.

After the initial analysis on the single gene level, we targeted candidate genes that had previously been associated with the development of hemispheric asymmetries: *LRRTM1* (*Francks et al., 2007*), *PCSK6* (*Scerri et al., 2011*; *Arning et al., 2013*; *Brandler et al., 2013*), *meiosis specific nuclear structural 1 (MNS1)* (*Brandler et al., 2013*), *polycystin 2, transient receptor potential cation channel (PKD2)* (*Brandler et al., 2013*), *AR* (*Arning et al., 2015*; *Hampson and Sankar, 2012*; *Medland et al., 2005*), *SET domain bifurcated 2 (SETDB2)* (*Ocklenburg et al., 2016*), and *catechol-O-methyltransferase (COMT)* (*Savitz et al., 2007*) have been reported as specific candidate genes for handedness, whereas *glutamate ionotropic receptor NMDA type subunit 2B (GRIN2B)* (*Ocklenburg et al., 2011b*), *FOXP2* (*Ocklenburg et al., 2013b*; *Pinel et al., 2012*), *KIAA031945* (*Pinel et al., 2012*), and *cholecystokinin A receptor (CCKAR)* (*Ocklenburg et al., 2013d*) have been associated with language lateralization. Other genes associated with left-right differentiation are *left-right determination factor 1 (LEFTY1)* (*Mittwoch, 2008*) and *nodal growth differentiation factor (NODAL)* (*Mittwoch, 2008*), *BDNF* (*Manns et al., 2008*), *LIM homeobox 1 (LHX1)* (*Tsang et al., 1999*), and *bone morphogenetic protein 7 (BMP7)* (*Abu-Khalil et al., 2004*). Among these, *AR* and *NODAL* were not expressed above the detection level at any developmental stage.

In order to identify functional groups of asymmetrically expressed genes, we used WebGestalt (RRID:SCR_006786) (*Wang et al., 2013*; *Zhang et al., 2005*) to carry out an enrichment analysis over all genes with a $\log_2$(fold change) > 1.5 per hemisphere and developmental stage with respect to GO groups (biological process, molecular function, cellular component).

## miRNA expression analysis and bioinformatics

The amount of RNA for miRNA sequencing was only sufficient in sample II, IV, and V (see *Supplementary file 1B*). The service 'Regulome sequencing' of GATC Biotech AG provided the analysis of miRNA and included the generation of a small- / miRNA-library for subsequent Illumina

single-read sequencing with 5 million reads as well as identification of common miRNAs with corresponding expression values (read counts). Originally, all samples yielded between 618 and 858 expressed miRNAs (sample II: 638 left, 858 right; sample IV: 838 left, 799 right; sample V: 618 left, 658 right). miRNAs with less than 10 read counts on both sides of the spinal cord were removed from analysis to prevent unrealistically high fold changes (*Hu et al., 2011*), which left 300 miRNA transcripts for sample II, 381 transcripts for sample IV, and 293 transcripts for sample V. In contrast to pure read counts, RPKM (reads per kilo base of exon model per million mapped reads) values correct for sequencing depth and gene length, so RPKM was calculated (RPKM = $[10^9 \times$ reads mapped to the transcript)/(total number of reads in the library $\times$ transcript length] [*Kazemian et al., 2015*]). Differential miRNA expression was defined as a minimum $\log_2$(fold change) of 1.5. Using Mirpath v3.0 (*Vlachos et al., 2015*), we identified genes that were likely (probability of interaction > 0.8) to be targeted by the asymmetrically expressed miRNA transcripts, which were then compared to the differentially expressed genes. Variance in asymmetrical gene expression was considered to be explained by miRNA if asymmetrically expressed genes were targets of asymmetrically expressed miRNAs. Additionally, we performed KEGG (RRID:SCR_012773) analyses (*Vlachos et al., 2015*) to identify biological pathways regulated by asymmetrically expressed miRNA transcripts.

## Methylation analysis and bioinformatics

DNA was bisulfite treated and adapter and primer sequences as well as bases with a phred quality score lower than Q15 were removed. After transformation into a C-to-T and G-to-A version, BIS-MARK (RRID:SCR_005604) (*Krueger and Andrews, 2011*) and Bowtie2 (*Langmead et al., 2009*) were applied on the sequence reads in order to align them to the *in silico* converted reference (*Homo sapiens*, hg19), which was refined using Bis-SNP (RRID:SCR_005439) (*Liu et al., 2012*) adopted GATK (RRID:SCR_001876) (*McKenna et al., 2010*; *DePristo et al., 2011*) modules. Bis-SNP then simultaneously determined the genotypes and methylation rates at each CpG site using Bayesian inference. DNA read statistics and methylation levels are reported in *Supplementary file 1F*. The annotation was performed using the UCSC genome browser (http://genome.ucsc.edu/.) for detected CpG sites and using the RefSeq database (http://www.ncbi.nlm.nih.gov/refseq/) for genes. Comparative methylation analysis (left vs. right) was performed by using Fishers exact test using the R-package methylKit (RRID:SCR_005177) (*Akalin et al., 2012*). P-values were adjusted for false discovery rate (FDR) using the SLIM method (*Wang et al., 2011*). CpG sites were considered as differentially methylated if FDR-corrected p-value was below 0.01 and % methylation difference between left and right was above 25%. For week 10 and 12, average % methylation difference was calculated for every CpG site that was asymmetrically methylated in both samples, whereas for week 14, only one sample was available. CpG sites were compared to asymmetrical gene expression data by matching chromosome positions. For each asymmetrically expressed gene, the number of hypermethylated CpG sites in the left and in the right spinal cord within this gene and 1500 nucleotides upstream was determined. A laterality quotient (LQ) was calculated [(right−left)/(right+left)*100] for each gene. Variance in asymmetrical gene expression was considered to be explained by differential methylation if methylation of CpG sites within one gene was strongly asymmetric (i.e. LQ > 25/LQ < −25) towards the opposite direction of gene expression.

## Acknowledgements

We thank the medical personnel at the gynecological clinic for their support and help.

## Additional information

### Funding

| Funder | Grant reference number | Author |
|---|---|---|
| Deutsche Forschungsgemeinschaft | Gu227/16-1 | Onur Güntürkün |

The funders had no role in study design, data collection and interpretation, or the decision to submit the work for publication.

## Author contributions
SO, Conceptualization, Formal analysis, Supervision, Writing—original draft, Project administration, Writing—review and editing; JS, Data curation, Software, Formal analysis, Visualization, Writing—original draft, Writing—review and editing; ZM, PF, Data curation, Investigation, Writing—review and editing; DM, Data curation, Investigation, Methodology, Writing—review and editing; RKl, SL, Formal analysis, Writing—review and editing; GK, MT, Methodology, Project administration, Writing—review and editing; CF, Formal analysis, Validation, Methodology, Writing—review and editing; JTE, Data curation, Validation, Investigation, Writing—review and editing; RKu, Conceptualization, Formal analysis, Investigation, Methodology, Writing—review and editing; OG, Conceptualization, Resources, Formal analysis, Supervision, Funding acquisition, Project administration, Writing—review and editing

## Author ORCIDs
Sebastian Ocklenburg, http://orcid.org/0000-0001-5882-3200

## Ethics
Human subjects: The study was approved by the Ethics Committee of the Medical Faculty of the Ruhr University Bochum (registration number 5056-14). All fetal tissue donors signed written informed consent.

# Additional files

## Supplementary files
• Supplementary file 1. (A) Description of tissue samples. (B) RNA quality measurements. (C) DNA quality measurements. (D) RNA read statistics. (E) DNA read statistics. (F) Methylation report. (G) Top 25 asymmetrically expressed genes per sample.

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
