## [Decision Letter]

Thank you for submitting your article "Epigenetic regulation of lateralized fetal spinal gene expression underlies hemispheric asymmetries" for consideration by *eLife*. Your article has been reviewed by three peer reviewers, and the evaluation has been overseen by a Reviewing Editor and Marianne Bronner as the Senior Editor. The following individuals involved in review of your submission have agreed to reveal their identity: Peter Hepper (Reviewer #2); Chris McManus (Reviewer #3).

The reviewers have discussed the reviews with one another and the Reviewing Editor has drafted this decision to help you prepare a revised submission.

Summary:

This paper studies asymmetry of gene expression in human fetal spinal cord. This is important as it allows for the demonstration that asymmetries are present before the spinal cord is connected to the cortex. The reviewers agreed that this was an exciting, important and novel paper.

Essential revisions:

1) There were originally six fetuses, the small number not being the fault of the researchers. Later there are only five and it took a while to find out what happened to the sixth one. The number of fetuses should also be stated in the Abstract, lest readers are potentially misled.

2) Figure 1 in particular has 2 fetuses at 8 and 10 weeks and only 1 at week 12. I wonder, given the very small Ns, and the novelty of the questions and the analyses whether perhaps it would make sense at 8 and 10 weeks to show the results separately for each of the two fetuses, as that would give readers some sense of the reliability of the results. Ditto perhaps for Figure 2 and Figure 3. It was nice though, in the supplementary file to see information for each fetus (but it was only in a footnote to [Supplementary-material SD7-data] that I found the fate of the sixth fetus).

3) I was not at all clear in the second paragraph of the subsection “Gene expression analysis and bioinformatics” how the bootstrapped t-tests were carried out. If there are two cases, with scores A and B, then the bootstrap can only come up with AA, AB or BB, with AB being the actual case. Is there any point in that? Or is the analysis somehow being done across all genes? More detail please. Sometimes, of course, it is also not really necessary to do statistics, and this might be one of those occasions.

---

## [Author Response]

*Essential revisions:*

*1) There were originally six fetuses, the small number not being the fault of the researchers. Later there are only five and it took a while to find out what happened to the sixth one. The number of fetuses should also be stated in the Abstract, lest readers are potentially misled.*

In accordance with the reviewers’ suggestions, we now added the number of fetuses to the Abstract. The failure of quality control for one sample that was not further processed is mentioned in the Methods section (Gene expression analysis and bioinformatics).

*2) Figure 1 in particular has 2 fetuses at 8 and 10 weeks and only 1 at week 12. I wonder, given the very small Ns, and the novelty of the questions and the analyses whether perhaps it would make sense at 8 and 10 weeks to show the results separately for each of the two fetuses, as that would give readers some sense of the reliability of the results. Ditto perhaps for Figure 2 and Figure 3. It was nice though, in the supplementary file to see information for each fetus (but it was only in a footnote to [Supplementary-material SD7-data] that I found the fate of the sixth fetus).*

We would prefer to retain the pooled expression analysis across both samples for the different timepoints for several reasons:

First, this approach has been used by previous studies on expression asymmetries (e.g. Sun et al., 2005; Karlebach & Francks, 2015) and essentially is standard in the field.

Especially given the low n, some control for sparse findings is essential. In the pooled analysis, only genes that showed reliable expression (FPKM > 1) in both the left and the right spinal cord of both fetuses were included in the analysis. By only analyzing genes that were reliably expressed in all four samples, we excluded sparse findings. Making the analysis on single fetus level potentially would lead to much more “noisy” data, which we would like to avoid.

Last, we doubt that doubling the number of figures in this manuscript that already has a large number of figures would enhance the manuscript’s readability.

We however see the adding more information on the single fetuses would benefit the manuscript and added a new [Supplementary-material SD7-data] showing the top 25 hits for each fetus separately.

We explicitly state in the Methods that the sixth fetus was not further analyzed due to quality control issues of the sample. The sentence reads: “Sample VI did not pass entry quality control and was not further processed.”

*3) I was not at all clear in the second paragraph of the subsection “Gene expression analysis and bioinformatics” how the bootstrapped t-tests were carried out. If there are two cases, with scores A and B, then the bootstrap can only come up with AA, AB or BB, with AB being the actual case. Is there any point in that? Or is the analysis somehow being done across all genes? More detail please. Sometimes, of course, it is also not really necessary to do statistics, and this might be one of those occasions.*

Bootstrapped t-tests were carried out as described by the reviewers (AA, AB or BB). We fully agree with the reviewer that making meaningful statistics with sample sizes this small is difficult and that instead relying on the biologically relevant fold changes might be a better representation of the data. Thus, in accordance with the reviewer’s suggestion, we now deleted these tests from the manuscript and completely rely on the fold changes for our results.